# Artificial Intelligence in Adaptive and Intelligent Educational System: A Review

**Jingwen Dong** , **Siti Nurulain Mohd Rum** *, **Khairul Azhar Kasmiran**, **Teh Noranis Mohd Aris** and **Raihani Mohamed**

Faculty of Computer Science & Information Technology, Universiti Putra Malaysia, Serdang 43400, Malaysia
* Correspondence: snurulain@upm.edu.my

**Abstract:** There has been much discussion among academics on how pupils may be taught online while yet maintaining a high degree of learning efficiency, in part because of the worldwide COVID-19 pandemic in the previous two years. Students may have trouble focusing due to a lack of teacher–student interaction, yet online learning has some advantages that are unavailable in traditional classrooms. The architecture of online courses for students is integrated into a system called the Adaptive and Intelligent Education System (AIES). In AIESs, reinforcement learning is often used in conjunction with the development of teaching strategies, and this reinforcement-learning-based system is known as RLATES. As a prerequisite to conducting research in this field, this paper undertakes the consolidation and analysis of existing research, design approaches, and model categories for adaptive and intelligent educational systems, with the hope of serving as a reference for scholars in the same field to help them gain access to the relevant information quickly and easily.

**Keywords:** distanced learning; adaptive and intelligent education system; reinforcement learning

## 1. Introduction

Machine learning technology can be used in many fields, such as commerce [1], biology [2], medicine [3], and pedagogy [4], where algorithms can be used to analyze data quickly and obtain results that are difficult for the human brain to calculate. In education, machine learning technology has also been widely used, for example, to analyze student performance and achievement and in order to implement the necessary measures to improve student satisfaction and help them graduate [5], and big data have been used to analyze data on students at different stages of learning in order to improve educational policy [6]. As the application of machine learning technology in the field of education continues to develop, scholars have started to use machine learning in order to develop teaching strategies, resulting in a series of educational systems, such as the Adaptive Learning System [7], Adaptive Intelligent Tutoring Systems [8], and the Adaptive and Intelligent Educational System [9]. In AIESs, reinforcement learning is often used in conjunction with the development of teaching strategies, and this reinforcement-learning-based system is known as RLATES [10].

If reinforcement learning is applied to the development of teaching strategies, the interaction between the student and the system can lead to a personalized optimal learning strategy [10]. Based on existing research, the performance of RLATES is often limited by the following factors:

1. Current research only applies the classical Q-learning algorithm to train the system and scheme appropriate teaching strategies [11], but the classical Q-learning algorithm has a weakness in that, in some cases, the Q-learning algorithm overestimates the value of the action [12].
2. The model in an intelligent teaching system may be unsatisfactory for cases that have not been repeatedly trained [13].

3. The over-practice problem means that more practice does not necessarily lead to a longer study time [14].
4. Next, let us further elaborate on these three limitations.

Firstly, with regard to the overestimation problem mentioned in the first point, which is mentioned in Van Hasselt's [12] article, this problem cannot be avoided by a classical Q-learning algorithm. Nevertheless, a large number of studies still choose to use a classical Q-learning algorithm when applying reinforcement learning to intelligent educational systems [9,13,15], and in these studies, the authors do not address the issue of overestimation. For the field of reinforcement learning, the double Q-learning and double DQN algorithms have been developed that are more sophisticated than the classical Q-learning algorithm, and potential overestimation problems can be avoided to some extent if these two algorithms are applied to the intelligent educational system.

Concerning item two, in Shawky and Badawi's [13] article, Q-learning is applied in an intelligent teaching system, but the model of the system suffers from an implementation problem if the user is not trained enough, i.e., if there is not a large number of repetitions for the system to learn, the system may not perform ideally. Despite the issue of the system, their contribution cannot be ignored. In their research, the system allows for new actions or states to be dynamically added to the system in real time, which means that teachers and students can independently add content to the system that they consider helpful or necessary, and the system can be automatically updated with new states or actions. This means that the system is fully adaptive and user-friendly.

About the last item, there are also authors who apply uncommon algorithms in instructional strategy development systems, such as Proximal Policy Optimization (PPO) [14] and the Partially Observable Markov Decision Process (POMDP) [16]. In Bassen's study, the over-practice problem was significant, as completing more activities did not always result in an increase in learners' engaged time. Nevertheless, the use of neural networks in this paper reduced the dimensionality of the state and action space, thus reducing the number of samples that the algorithm needed to converge. However, in Zhang's research, the over-practice problem did not exist, but the reward values of the reinforcement learning algorithm in the paper could be adjusted to obtain better performance. The most striking contribution of this study is that the Partially Observable Markov Decision Process can still be used when the information provided by the student is incomplete, as the method allows for the optimal solution to still be given in such a case. Moreover, the system proposed in Zhang's paper can obtain the information locally when providing a correct answer to a question from student users.

The purpose of this paper is to provide a quick overview of how reinforcement learning can be applied in AIESs and to compare and summarize related works so far to assist scholars in related fields develop their works better.

This paper contains four sections. Section 2 introduces different types of reinforcement learning and some related works and the principle of reinforcement learning; moreover, different reinforcement learning algorithms are introduced, and a comparison between reinforcement learning and Bayesian networks is presented at the end. Section 3 presents the systematic structure of RLATES and comparison among some related studies. Finally, Section 4 presents the conclusions and limitations.

## 2. Reinforcement Learning

Reinforcement learning is a branch of machine learning, a parallel paradigm to supervised and unsupervised learning. Figure 1 clearly shows the machine learning paradigms.

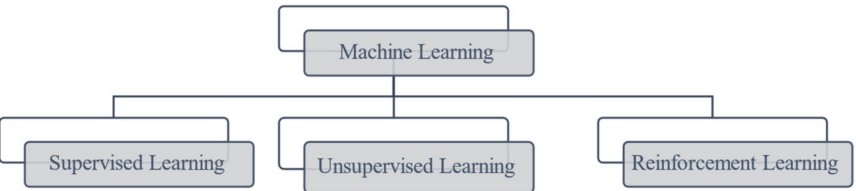

**Figure 1.** Machine Learning paradigms.

Reinforcement learning generally consists of five elements [17], namely, agent, environment, action, reward, and state. In a reinforcement learning algorithm, two main elements interact, the agent and the environment. The agent can interact with the environment to generate actions, and the environment returns a reward based on the actions of the agent; then, the agent can perceive the state of the environment and perform the next action based on the reward received. Figure 2 better explains this process.

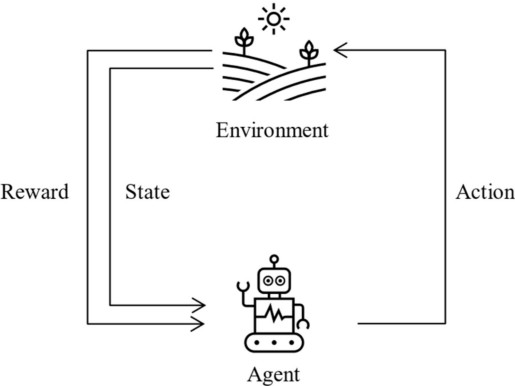

**Figure 2.** A typical framework of RL scenario.

### 2.1. Model-Based and Model-Free Reinforcement Learning

Reinforcement learning algorithms can be divided into two broad categories, model-based and model-free reinforcement learning algorithms [18]. Simplistically, model-based reinforcement learning algorithms require the agent to learn in the environment, combine the learning experience to generate a model, and then formulate an action strategy based on this model; in this case, the agent interacts with the virtual environment. For model-free reinforcement learning algorithms, the agent learns and formulates action strategies directly from the experience of interacting with the environment, in which case no model is generated, and the agent interacts with the real environment.

Model-based reinforcement learning algorithms are not used as much as model-free reinforcement learning algorithms in the field of pedagogy, and the data in Table 1 were obtained by examining the articles published in the mainstream databases over the last five years (2018–2022).

**Table 1.** Number of articles for model-based and model-free reinforcement learning algorithm.

| Database | Model-Free | Model-Based |
|---|---|---|
| SpringerLink | 93 | 99 |
| ScienceDirect | 122 | 81 |
| IEEE Xplore | 25 | 27 |
| Total | 240 | 207 |

Many algorithms, such as Imagination-Augmented Agents (I2As) [19], World Models [20], and Model-Based Value Expansion (MBEV) [21], belong to the category of model-based reinforcement learning algorithms.

Compared with model-based reinforcement learning algorithms, model-free reinforcement learning algorithms, such as Q-learning [22], Deep Q-Network (DQN) [23], Deep Deterministic Policy Gradient (DDPG) [24], Categorical Distributional RL (C51) [25], Proximal Policy Optimization (PPO) [26], and Soft Actor-Critic (SAC) [27], are applied in a broader range.

Figures 3 and 4 provide a more intuitive description of the classification for reinforcement learning algorithms.

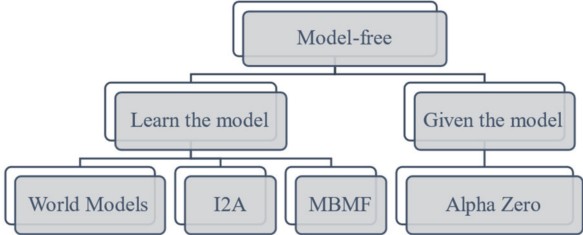

**Figure 3.** Model-free reinforcement learning algorithms.

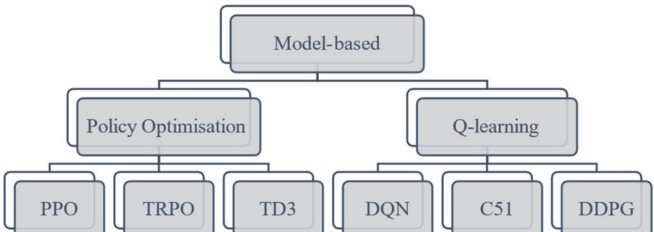

**Figure 4.** Model-based reinforcement learning algorithms.

## 2.2. Markov Decision Processes

Reinforcement learning is established based on a Markov Decision Process (MDP), which is a tuple consisting of a finite set of actions and state transfer probabilities. The MDP model comprises a reward function and a transition function; the functions are shown below [28].

$$T : S \times A \times S \rightarrow [0,1] \tag{1}$$

$$R : S \times A \times S \rightarrow \mathbb{R} \tag{2}$$

Many different versions of Markov decision processes have evolved over time, and three types of Markov decision processes are succinctly described in Table 2.

**Table 2.** Different versions of Markov decision processes.

| MDP Versions | System Form | State Characteristic | Reference |
|---|---|---|---|
| Fully Observable MDP (FOMDP) | Discrete | All observable | [29] |
| Partially Observable MDP (POMDP) | Discrete | Partially observable | [16] |
| Semi-MDP(SMDP) | Continuous | All observable | [30] |

## 2.3. Q-Learning

The quintessential reinforcement learning algorithm is Q-learning, a model-free, off-policy algorithm, which is also one of the most commonly used in many reinforcement-learning-related studies.

The essence of Q-learning is derived from the Bellman equation [31], which is formulated as shown below.

$$V_\pi(s) = \sum_a \pi(s,a) \sum_{s'} p\left(s' \mid s,a\right) \left(\mathbb{W}_{s \rightarrow s'|a} + \gamma V_\pi\left(s'\right)\right) \tag{3}$$

$$Q_\pi(s,a) = \sum_{s'} p(s' \mid s,a) \left( \&\mathbb{W}_{s\to s'|a} + \gamma \sum_{s'} \pi(s',a') Q_\pi(s',a') \right) \tag{4}$$

In the Bellman equation,

$$\mathbb{W}_{s\to s'|a} = \mathbb{E}\left[ r_{t+1} \mid s_t = s, a_t = a, s_{t+1} = s' \right] \tag{5}$$

$V_\pi$ (s) is the state value function, and $Q_\pi$ (s, a) is the action value function. In the equation above, *s* represents the state, as the state *s* to the next state *s* + 1 is uncertain; hence, the expectation E needs to be added to the equation, and *r* represents the reward.

For the Q-learning algorithm, the policy is selected based on the Q-table, which is structured as S*A, where S represents the state, and A represents the action. Based on the given Q-table, the next action can be determined based on the current state of the environment. After deciding which action to perform, the agent proceeds with the next action accordingly, and after performing the action, the agent receives a reward from the environment. After each action, the Q-table in the environment is updated, and the adjustment of the Q-table is carried out according to the following formula [32]:

$$Q(s_t,a_t) \leftarrow Q(s_t,a_t) + \alpha \left[ r + \gamma \max_{a_t} Q(s_{t+1},a_t) - Q(s_t,a_t) \right] \tag{6}$$

In this formula, *s* represents the state, *a* represents the action, *r* represents the reward, $\alpha$ represents the learning rate, and $\gamma$ represents the discount factor. $\alpha$ and $\gamma$ both range on a scale from 0 to 1.

However, in 1993, Thrun and Schwartz [33] suggested that, since random errors may uniformly occur in the action values, there may be a bias in the path of progressively seeking the optimal, consequently resulting in a sub-optimal solution rather than the optimal solution. Seventeen years later, van Hasselt's paper [34] showed that environmental noise might also cause overestimation problems in classical Q-learning algorithms.

In Hasselt's paper, a Double Q-learning algorithm was proposed to solve the overestimation problem, and later, Van Hasselt, Guez, and Silver [12] also proposed an algorithm called Double DQN to solve this problem; the details of which are presented in the next section.

### 2.4. Deep Q-Network and Double Deep Q-Network

DQN is a subordinate branch of deep reinforcement learning and is a value-based reinforcement learning algorithm [23], in which only the value function and not the policy network is involved. It is an upgrade to classical Q-learning algorithms. As tasks become increasingly complicated, classical Q-learning algorithms do not sufficiently perform these tasks because in classical Q-learning algorithms, if the Q-table is excessively huge, it is a challenge for the computer to store all the data, thus making computation difficult and an excessively large Q-table also makes the retrieval of Q-values sluggish. When convolutional neural networks are implemented, states and actions can be used as inputs to calculate the corresponding Q-values, thus sparing the space required to store the Q-table and the time required to retrieve the Q-values. DQN introduces a convolutional neural network as an approximation function to approximate the value function. The ε-greedy strategy is used in DQN to select the next action. ε-greedy is a greedy strategy, where the option with the highest reward is chosen for each selection.

Next, the operation of the DQN is explained. First, the environment provides a state value, and the agent can determine all the states and actions currently available to the agent based on the value function. In order to determine the next action, an ε-greedy policy is used to determine what the next action will be, and after the selection is completed, the environment returns the corresponding reward value and the new state value. Up to this stage, it is a selection loop, and DQN makes a continuous loop of these to obtain the optimal policy [21].

Unfortunately, similarly to Q-learning, the overestimation problem cannot be avoided by DQN, but an article published in 2016 proposed a new algorithm that provides an initial solution to the overestimation problem [12]. In this article, a function called Double DQN is defined, which solves the overestimation problem at its core by stopping the constant propagation of overestimation between states. In simple terms, after selecting the maximum Q-value for the first time, the maximum Q-value is picked again in another network according to the corresponding action value, and if the Q-value is the maximum value in both networks, then the probability of overestimation is reduced to a large extent, even though it still occurs sometimes. Double DQN is a significant improvement over classical Q-learning and classical DQN for the overestimation problem.

*2.5. Comparison with Bayesian Network*

For intelligent educational systems, there are studies that apply other artificial intelligence algorithms to systems in addition to reinforcement learning, such as the application of the Bayesian Network to intelligent teaching systems [35,36].

Similar to reinforcement learning, Bayesian Network is also a process of seeking optimal solutions. If applied to intelligent educational systems, the system design is largely consistent with the concept of learning from students' knowledge and student characteristics and, based on this, recommends appropriate teaching strategies for the corresponding groups.

But essentially, Bayesian Network is an iterative process and is stateless [37]. Although the process is called iterative, each function call is independent of others, which means that the previous call does not affect the next step.

However, for reinforcement learning, the optimization process is an overall process that is stateful, and each transition between states affects each other, which means that the choice of the previous step affects the transition to the next state. If the optimum solution is desired by reinforcement learning, then the accumulative sum of all rewards is required, which is quite different from Bayesian Network.

## 3. Reinforcement Learning in Adaptive and Intelligent Educational System (RLATES)

Distance learning has attracted increasing attention in recent years. When students and teachers are unable to attend classes face to face in the same classroom, distance learning is essential. Sometimes, learning through resources on the web (text, video, pictures, voice, etc.) or online tutorials provided by teachers can fulfill the basic requirements of distance learning. However, when questions are encountered, students are unable to accurately find the answers through online resources, and the one-to-many teacher–student sessions ultimately do not fulfill one function, which is adaptive instructions [38]. The reason why one-to-one lessons are more effective and more satisfying than small group lessons is that one-to-one lessons allow for personalized teaching strategies, but because of the high financial cost of one-to-one lessons, it is impossible to extend this approach to all groups of students. Based on this situation, the Adaptive Intelligence Educational System was designed to allow students to develop their own personalized teaching strategies if they have access to a computer, thus allowing them to benefit from a one-to-one teaching model at a relatively low cost and allowing each student to have their own virtual teacher.

The principle of AIESs is to resequence all course knowledge modules based on student characteristics, and a variety of machine learning techniques are used in the system to learn student characteristics [39]. According to the principles of AIESs, if reinforcement learning is introduced into the system, it allows the student to interact with the system and allows the system model to continuously improve its learning, thus enhancing its performance. The type of system that introduces reinforcement learning into AIESs is called RLATES.

RLATES comprises two models, the knowledge model and the pedagogical strategy model [11]. In the knowledge model, the content of the teaching is decided, for example, which chapters of the textbook will be covered and which format (video, audio, text, or

pictures) will be used for delivery. In the pedagogical strategy model, the teaching strategy is developed, which determines how the material will be delivered.

However, RLATES is not available for teaching directly from the beginning. At the early stage, the model needs to first be trained by feeding it with training data so that the system learns which teaching strategy to use when confronting students with different characteristics. Therefore, the whole experimental process should be separated into two phases when designing the system, the training phase and the teaching phase [10]. Only after the model has been successfully trained can it be implemented into real teaching.

### 3.1. Current Research

In this section, the current status of the research in the domain of intelligent educational systems is presented. Although there are numerous studies that focus on intelligent educational systems similar to AIESs, the retrieval shows that only a fraction of them adopted reinforcement learning algorithms. The details are shown in Table 3.

**Table 3.** Current research for intelligent tutoring system.

| No. | Author and Year | Algorithm | Assessment Metrics | Description |
|---|---|---|---|---|
| 1 | Dorça et al. [9] | Q-learning | Performance value (PFM)/distance between learning style (DLS) | Three different automated control strategies are proposed to detect and learn from students' learning styles. |
| 2 | Iglesias et al. [10] | Q-learning | Number of actions/number of students | Apply the RL in AIESs with database design. |
| 3 | Iglesias et al. [11] | Q-learning | Time consumption/number of students | Apply the reinforcement learning algorithm in AIESs. |
| 4 | Shawky and Badawi [13] | Q-learning | Number of actions/number of steps/cumulative rewards | The system can update states and adding new states or actions automatically. |
| 5 | Thomaz and Breazeal [15] | Q-learning | Number of actions/state/trials | Modify the RL algorithm to separate guidance and feedback channel. |
| 6 | Bassen et al., 2020 [14] | PPO (Proximal Policy Optimization) | Course completion rate/learning gains | Applied neural network to reduce number of samples required for the algorithm to converge. |
| 7 | Zhang, 2013 [16] | POMDP (Partially Observable Markov Decision Process) | Rejection rate | The model provides local access to the information when selecting the correct answer to a student's question. |

According to Table 3, it can be concluded that, in the domain of intelligent teaching systems, most authors still adopt the classical Q-learning algorithm since the Q-learning algorithm is a model-free and policy-free reinforcement learning algorithm that is suitable for implementation in intelligent teaching systems. However, due to the defects of Q-learning, the processing speed is sluggish, and the system response time increases when the Q-table is excessively large. Nonetheless, the Q-learning algorithm is one of the classical algorithms of reinforcement learning and is relatively simple in practical applications compared with other model-free reinforcement learning algorithms, which is probably part of the reason why many authors chose the Q-learning algorithm in their studies.

For the articles listed above, although articles 1–5 all adopt the Q-learning algorithm, their evaluation metrics are distinct. Most of the authors selected the number of actions, the number of students, or time consumption to evaluate the performance of the model. However, the most remarkable article is article 1, in which the authors develop an evaluation metric themselves called PFM. According to the authors' settings in that article, if $PFM \geq 60$, then the performance of the model is good, and if $PFM < 60$, then the performance of the model is poor. Meanwhile, an assessment using PFM can also indicate the difficulty of the learning content to some extent; if the performance is poor, then this indicates that the learning content is probably relatively difficult. The authors use this evaluation metric to compare the three strategies within the article, and although it does

not permit a horizontal comparison of the model's performance to other articles, it makes the evaluation results more intuitive and straightforward to peruse and understand.

### 3.2. Applied Reinforcement Learning in RLATES

Based on the introduction to reinforcement learning in Section 3, it can be seen that reinforcement learning comprises five main components. In order to apply reinforcement learning to RLATES, it is essential to ensure that the components of the system correspond to each of the five components of reinforcement learning algorithms. Therefore, in this section, the application of reinforcement learning to RLATES is introduced.

First, the following descriptions are given of how the components of RLATES correspond to those of the reinforcement learning algorithm [10]:

1.  Agent: In RLATES, the agent refers to the student. The learning system is used through the student interacting with the system for subsequent processes; therefore, the student corresponds to the agent in the reinforcement learning algorithm.
2.  Environment: In a broad sense, the environment is the entire knowledge structure of the system, and it collects information on the characteristics of the students and tests their knowledge through exams and quizzes distributed throughout the knowledge modules.
3.  Action: Actions are the selections that an agent needs to take at each step, so in RLATES, the actions correspond to the knowledge modules, each of which represents an action.
4.  State: In reinforcement learning algorithms, the state refers to the state that the environment returns to when the agent performs an action. Therefore, in RLATES, the state corresponds to the student's learning state, i.e., how the student mastered the knowledge. Here, a vector is used to store the data, and all state values are in the range of 0–1. For a student, if the knowledge has been fully mastered and correctly understood, the state value is set at 1. If the knowledge has not been mastered by the student, then the state value is set at 0.
5.  Reward: For reinforcement learning algorithms, each selection returns a different reward value, and similarly, in RLATES, each knowledge module corresponds to a different reward according to the significance. Moreover, in RLATES, the intention is to maximize the cumulative value of this reward.

Next, the application of the reinforcement learning algorithm to RLATES is described in Algorithm 1. Coupling the components in RLATES to the elements in the reinforcement learning algorithm yields the following process [10,11]:

---

**Algorithm 1** Apply reinforcement learning algorithm to RLATES

---

Initialize $Q(s, a)$ for $s \in S$ and $a \in A$
Test the current situation of student's knowledge $s$
Loop for each episode,
Pick a knowledge module $a$, show this module to the student, by using the $\varepsilon$-greedy policy
Get the reward $r$, while the RLATES goal is achieved, a positive $r$ will be obtained, else a null $r$ will be obtained.
Test the current situation of student's knowledge $s'$
Update $Q(s, a)$:
$$Q(s, a) \leftarrow Q(s, a) + \alpha \left[ R + \gamma \max_a Q(s', a) - Q(s, a) \right]$$
$s \leftarrow s'$
until $s$ reaches the goal state

---

## 4. Conclusions

In this paper, a literature review on the development of adaptive intelligent educational systems is provided, and work related to the application of reinforcement learning to intelligent educational systems is also discussed, as well as a brief introduction to the principles of the systems and algorithms. This paper synthesizes the research work conducted

in recent years and can assist researchers in related domains in developing their work. The following conclusions can be extracted from this literature review:

1.  Due to the features of intelligent educational systems, reinforcement learning is appropriate for application in the construction of a system and can be very helpful in providing proper teaching strategies for students with the same characteristics.
2.  Although many scholars have worked on how to integrate reinforcement learning into intelligent instructional systems, the majority still only adopt classical Q-learning algorithms, and the application of the more sophisticated reinforcement learning algorithms to the field of intelligent instructional systems has rarely been conducted. When evaluating the system performance, most studies use similar evaluation metrics, which facilitates scholars to make comparisons between different studies. However, some studies have developed their own evaluation metrics, which can better evaluate the experimental results for future optimization, but the disadvantage is that they cannot compare the experimental results with other studies, which has limitations.
3.  Research on the application of reinforcement learning to intelligent instructional systems has rarely been conducted in recent years, but as online learning is increasingly required by students, research in this area is expected to increase in popularity in the future. Although online education cannot completely replace offline education, the combination of computer technology and education can make education gradually online, which is the trend of future development.

Although both reinforcement learning and AIESs are presented and analyzed as extensively as possible in this paper, these are based on literature aspects only and have not been validated and analyzed in practical experiments, which is a limitation of this paper.

In future work, we plan to combine the Bayesian Network with parts of the reinforcement learning process in order to improve the computational efficiency of the algorithm and the working efficiency of the system.

**Author Contributions:** Conceptualization, all the authors; writing—original draft preparation, J.D.; writing—review and editing, all the authors; supervision, S.N.M.R., K.A.K., T.N.M.A. and R.M.; All authors have read and agreed to the published version of the manuscript.

**Funding:** This research received no external funding.

**Data Availability Statement:** Not applicable.

**Conflicts of Interest:** The authors declare no conflict of interest.

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
