# Peer review of "Artificial Intelligence in Adaptive and Intelligent Educational System: A Review"

_futureinternet, doi:10.3390/fi14090245_

Round 1
Reviewer 1 Report
Reviewer's summary after reading the manuscript:
There has been much discussion among academics on how pupils may be taught online while yet maintaining a high degree of learning efficiency, in part because of the worldwide COVID-19 pandemic in the previous two years. Students may have trouble focusing due to a lack of teacher-student interaction, yet online learning has some advantages that are unavailable in traditional classrooms. The Adaptive and Intelligent Education System (AIES) is a system that incorporates the design of students' online courses. By applying algorithms from reinforcement learning, the system may learn the learning characteristics of the students over time through interaction with the system environment, and then modify the content's presentation order to better suit the needs of students with those qualities. As a prerequisite to conducting research in this field, this paper undertakes the consolidation and analysis of existing research, design approaches, and model categories for adaptive and intelligent educational systems, with the hope of serving as a reference for scholars in the same field to help them gain access to the relevant information quickly and easily.
----------------------------------------
Dear authors, thank you for your manuscript. I enjoyed reading it. Presented are some suggestions to improve it:
(1) Please consider modifying the title of the manuscript to include the words "Artificial Intelligence" before the words "Reinforcement Learning" so that it would be easier for potential readers to find your study. It also helps educators who may not be so familiar with this technical term of "reinforcement learning" to understand that it is related to AI and not to educational psychology or pedagogy.
(2) Please consider including Bayesian Network (BN) based probabilistic approaches in your literature review, and how similar or different they are from RL-based approaches because BN is the 'AI engine' powering many Adaptive and Intelligent Education Systems (AIES) being commercially offered in the education industry.
(3) Please include a "Limitations" section to discuss what were the challenges faced, and how your team overcame those challenges. This would be very beneficial to the readers as they would be able to learn from your expert knowledge.
(4) To improve the impact and readership of your manuscript, the authors need to clearly articulate in the Abstract and in the Introduction sections about the uniqueness or novelty of this article, and why or how it is different from other similar articles. The Conclusion section is too short. Can the authors please elaborate more about how this study is relevant to "the future of the development of the Internet" since it was submitted for publication in the journal "Future Internet"?
(5) Please substantially expand your review work.
(6) Many of the references cited are not yet properly formatted according to MDPI's guidelines. For example, the DOIs of many of the journal papers cited are not included yet. For the references, instead of formatting "by-hand", please kindly consider using the free Zotero software (https://www.zotero.org/), and select "Multidisciplinary Digital Publishing Institute" as the citation format, since there are currently 38 citations in your manuscript, and there may probably be more once you have revised the manuscript.
(7) There are some easily-missed typos throughout the manuscript. For example, on page 8 of the manuscript, it should be "3.2 Applied reinforcement learning in RLATES" instead of '3.2 Apply reinforcement learning in RLATES'.
(8) Please kindly consider engaging the professional editorial services of an English language proofreader to rectify the typo errors, and improve the flow of the manuscript.
Thank you.
Reviewer 2 Report
Interesting topic. The paper is not about research, but a bibliographic survey on the state of research in the topic. The paper can be accepted, but it should be required to go under some (just involving format, not ideas) re-writting and restructure, in order to improve the impact of its message. What follows are some suggestions in this direction:
>> The abstract is rather an introduction than an abstract. Actually, the abstract itself could be just (all sentences are from the paper, but the selection of the sentences is of the reviewer):
“As an application of machine learning technology in the field of education, the architecture of online courses for students is integrated into a system called the Adaptive and Intelligent Education System (AIES).In AIES, reinforcement learning is often used in conjunction with the development of teaching strategies and this reinforcement-learning-based system is known as RLATES. In this paper, the consolidation and analysis of existing research, design approaches, and model categories for adaptive and intelligent educational systems are carried out as a pre-requisite to conducting research in this field, aiming to provide a reference for scholars in the same field to help them gain valuable information conveniently and efficiently.”
since the remaining information now appearing in the Abstract is just about context, motivation, etc.
>>> page 2: it is written
- Current research only applies the classical Q-learning algorithm to train the system and scheme appropriate teaching strategies, but the classical Q-learning algorithm has a weakness in that, in some cases, the Q-learning algorithm overestimates the value of the action [12].
and a few lies later,
- However, an analysis of existing research shows that most systems use the classical Q-learning algorithm [10,15,16], but the classical Q-learning algorithm has a drawback, namely, the overestimation problem [12],
Obviously, the overestimation problem does no need to be introduced twice almost with the same words…. The same happens for the repeatedly trained problem, and the over-practice problem. I guess the authors want to include the three main problems summarily, as items 1, 2, and 3 (page 2) and then they want to develop in more detail this summary description. But such explanation is not clear, maybe the sentence
“However, an analysis of existing research shows…”
should be written after declaring:
“Let us describe this three items in detail. Concerning item 1), an analysis of the existing research shows….”
(or something like this).
>> Section 1, Introduction, should contain, at the end, some description of the goals of the paper, and its structure, etc. something like what is now written in the Conclusion
“In this paper, a literature review on the development of adaptive intelligent educational systems is provided and work related to the application of reinforcement learning to intelligent educational systems is also discussed, as well as a brief introduction to the principles of the systems and algorithms. This paper synthesizes the research work conducted in recent years and can assist researchers in related domains to develop their work.”
>> Right now Section 1 just introduces some topics that are relevant for the paper (without mentioning their role in the paper), but other topics, also relevant for the paper, are in Section 2, etc. Please restructure… and perhaps consider writting a true introduction that describes the context, the goals of the paper (something as the current Abstract, extended)…And then the current content of Sections 1, 2 and 3 could have independent relevance…without having the reader waiting for their relation to the paper until subsection 3.1!
>> Perhaps the (interesting) Conclusions (items 1, 2, 3) should be argued and expanded a little bit more.
